# Amitotic Cell Division, Malignancy, and Resistance to Anticancer Agents: A Tribute to Drs. Walen and Rajaraman

**DOI:** 10.3390/cancers16173106

**Published:** 2024-09-08

**Authors:** Razmik Mirzayans, David Murray

**Affiliations:** Department of Oncology, Cross Cancer Institute, University of Alberta, Edmonton, AB T6G 1Z2, Canada; david.murray5@ahs.ca

**Keywords:** polyploidy, amitotic cell division, neosis, genome chaos, polyploid giant cancer cells, intratumor heterogeneity, therapy resistance

## Abstract

**Simple Summary:**

Human cells are typically diploid, that is, they contain two sets of chromosomes. Some cells, however, contain three or more sets of chromosomes and are called polyploid. In the early 2000s, Kirsten Walen and Rengaswami Rajaraman and his associates independently reported that polyploid human cells are capable of undergoing asymmetric division via nuclear budding and bursting (amitosis), giving rise to daughter cells that proliferate rapidly and can contribute to tumorigenesis (transformation of normal cells to a neoplastic state). This commentary provides an update on these intriguing discoveries.

**Abstract:**

Cell division is crucial for the survival of living organisms. Human cells undergo three types of cell division: mitosis, meiosis, and amitosis. The former two types occur in somatic cells and germ cells, respectively. Amitosis involves nuclear budding and occurs in cells that exhibit abnormal nuclear morphology (e.g., polyploidy) with increased cell size. In the early 2000s, Kirsten Walen and Rengaswami Rajaraman and his associates independently reported that polyploid human cells are capable of producing progeny via amitotic cell division, and that a subset of emerging daughter cells proliferate rapidly, exhibit stem cell-like properties, and can contribute to tumorigenesis. Polyploid cells that arise in solid tumors/tumor-derived cell lines are referred to as polyploid giant cancer cells (PGCCs) and are known to contribute to therapy resistance and disease recurrence following anticancer treatment. This commentary provides an update on some of these intriguing discoveries as a tribute to Drs. Walen and Rajaraman.

## 1. Introduction

### 1.1. Amitotic Cell Division in Polyploid Mammalian Cells

In 2020, the science writer Kerry Benson published a news blog on cancer cell polyploidy highlighting the observations reported by a team at Brown University [1]. Kerry stated that “as researchers and medical professionals work to develop new treatments for cancer, they face a variety of challenges. One is intratumor heterogeneity—the presence of multiple kinds of cancer cells within the same tumor. Often, these “mosaic” tumors include cells, such as polyploidal giant cancer cells, that have evolved to become aggressive and resistant to chemotherapy and radiation.”

In the past, polyploid giant cancer cells (PGCCs) have been largely ignored, said Kerry [1], because PGCCs fail to undergo mitosis, a highly regulated process by which the nuclear and cytoplasmic components of a mother cell are divided into two daughter cells. However, recent studies reported by Xuan et al. [2] have demonstrated that such giant cells can produce rapidly proliferating progeny via amitotic cell division (nuclear budding and bursting similar to simple organisms like fungi). “These cells (emerging from PGCCs) appear to play an active role in invasion and metastasis, so targeting their migratory persistence could limit their effects on cancer progression” noted Michelle Dawson, the senior author of the study [2].

The observation that polyploid human cells can undergo amitotic cell division is not new. In the early 2000s, for example, seminal reports published by Kirsten Walen [3] and Rengaswami Rajaraman and associates (Sundaram et al. [4]) underscored the significance of amitotic cell division of polyploid human cells in the context of tumorigenesis and resistance to anticancer agents. 

### 1.2. Objectives

The purpose of this commentary is to pay tribute to Drs. Walen [3,5,6,7,8,9,10,11] and Rajaraman [4,12,13,14] for their contributions to our understanding of cancer biology in general, and cancer cell repopulation following radio/chemotherapy in particular.

Specifically, this article is arranged as follows: (i) role of polyploidy-amitosis in tumorigenesis (Section 2); (ii) fate of cancer cells with genome instability (Section 3, Section 4, Section 5 and Section 6); and (iii) genome instability in human hereditary cancer-prone disorders, with a focus on Li-Fraumeni syndrome (Section 7).

## 2. Polyploidy, Amitosis, and In Vitro Cell Transformation

Huang and Zhou have recently published a comprehensive review entitled “DNA damage repair: historical perspectives, mechanistic pathways and clinical translation for targeted cancer therapy” [15]. They chronicled the history of the DNA damage and repair journey from the landmark discovery of gene mutation in 1927, the importance of DNA repair pathways in preventing gene mutations and cancer in the 1960s, defining apoptosis in 1972, and the identification of oncogenes, cell cycle checkpoints, and the DNA damage response since the 1980s (also see Figure 1). These discoveries led to the somatic mutation theory of cancer and an approach to cancer therapy purely based on the reductionist theory.

Reductionism is centered on the idea that complex diseases such as cancer can be better explained by breaking them down (“reducing”) into small, simple pieces and studying each piece separately. Accordingly, many groups, our own included (e.g., [16,17,18,19,20]), devoted considerable efforts to characterize signaling pathways that are altered in cancer, to identify the many (hundreds and thousands) of factors that play roles in each pathway (e.g., in the p53 “fireworks” [21,22]), and to conceptualize how these various pathways interact to influence cell fate. The danger of relying on such information-generating approaches to cancer/cancer therapy (high content screens, omics, etc.) has become apparent in the past decade, as discussed by eminent scientists such as Robert Weinberg in 2014 (“Coming Full Circle—From Endless Complexity to Simplicity and Back Again” [23]), Nobel Prize Laureate William Kailen in 2017 (“Publish Houses of Brick, not Mansions of Straw” [24]), Henry Heng in 2019 (“Genome Chaos: Rethinking Genetics, Evolution, and Molecular Medicine” [24], and others (reviewed in [25,26]).

Some observations reported since the 1990s did not support the somatic mutation theory of cancer (e.g., [27] and references therein), which raised the question as to whether mutations beget cancers or cancers beget mutations [27]. To this end, in the early-/mid-2000s Kirsten Walen reported a series of studies demonstrating a non-mutational basis for the transformation of some human cells grown in culture [3,5,6,7]. The experiments were performed with various cell types, including human diploid epithelial cells (i.e., amniocytes) [3,5] and dermal fibroblasts [6,7] that were passaged until they entered the replicative senescence state. Transformation occurred in these cultures either spontaneously or after infection with SV40. The following sequential events in the process of transformation were reported: (i) a small proportion (1–3%) of cells within senescent cultures increased their nuclear contents through endoreduplication; (ii) amitosis/fragmentation of the resulting polyploid nuclei gave rise to multinucleated giant cells; (iii) individual nuclei became surrounded by a cell membrane and exited the multinucleated mother cells via budding; (iv) these daughter cells exhibited extended life span in culture (immortalization) and indicators of neoplastic transformation based on cytogenetic analysis. (The proportion of daughter cells, emerging from each giant cell, that showed these properties was not determined).

Walen’s work continued to shed light on the complexity of tumor initiation and progression that extends far beyond the canonical somatic mutation theory of cancer. In her 2014 article, for example, Walen highlighted the occurrence of parental genome segregation or “gonomery” (a well-known 1/2-size cell volume reduction process in plants) in senescent human polyploid fibroblasts [8]. Gonomery-based functional division in fibroblasts generated near-diploid daughter cells, which showed of a proliferative advantage over the cells of origin. The article was entitled “Neoplastic-Like Cell Changes of Normal Fibroblast Cells Associated with Evolutionary Conserved Maternal and Paternal Genomic Autonomous Behavior…” [8].

As a continuation of this work, Salmina et al. [28] determined how tumor cells segregate their replicated parental genomes which reunite to undergo a meiosis-like recombination after the next replication cycle. (This study was led by Jekaterina Erenpreisa, a long-time colleague of Dr. Walen.) The authors reported that “pseudo-mitosis commonly occurs with a mitotically disabled spindle and closed telomeres, thereby evading practically all of the checkpoints of the mitotic cell cycle. Parental genome segregations with spindle uncoupling has been suggested by Walen in normal human senescing fibroblast cultures and in those treated with spindle poisons, and extrapolated to stem cell biology, highlighting gonomery…as an operating unit of genome segregations” and a mechanism of malignant transformation [28].

In her most recent articles [10,11], Walen discussed her discoveries in relation to epigenetic reprogramming, atavistic mechanisms, and “The First Cell” model of cancer [29]. In her 2021 review [10], Walen stated that “Three-four years ago scientists on the television screen promised cancer-eradication in a 5–10 year time-frame. But now 5 years have gone and the promise of eradication is no longer heard. Why?”

This is an important question. What are the reasons for continuing failures in cancer therapy despite many promises of modern therapeutic strategies (e.g., targeted cancer therapy, precision oncology, synthetic “lethality”) for decades? Perhaps “The assumption of normal mitosis in cancer proliferation is wrong!” said Walen [11]. Other reasons why “The ‘War on Cancer’ Isn’t Yet Won” [30] have been discussed [22,26,31,32,33].

## 3. Fate of Cancer Cells with Genome Instability

### 3.1. Amitosis in Solid Tumors/Tumor-Derived Cell Lines

Single cell biology has revealed that solid tumors and tumor-derived cell lines contain a subset of cancer cells with extensive genome instability that are readily distinguishable from the bulk of cancer cells by virtue of their enormous size (reviewed in, e.g., [34,35,36,37,38,39,40,41]). A variety of terms have been used to describe such large-sized cells, including PGCCs, polyaneuploid cancer cells, osteoclast-like cancer cells, pleomorphic cancer cells, blastomere cancer cells, and multinucleated cancer cells [41]. For simplicity, we will refer to giant cells with a highly enlarged nucleus, multiple nuclei, and/or multiple micronuclei as PGCCs.

Key discoveries reported between 1956 and the mid-2000s, which have been particularly useful for our group to understand the significance of PGCCs in tumorigenesis and therapy resistance of solid tumors (see, e.g., [42,43,44]), are pointed out below. Figure 2 is reproduced from our 2008 review [42].

In 1956, Puck and Marcus demonstrated that exposure of HeLa (cervical carcinoma) cells to ionizing radiation results in the creation of multinucleated giant cells that fail to form macroscopic colonies (aggregates of at least 50 cells) within ~10 days post-irradiation [45] (also see Appendix A). These giant cells were shown to remain adherent to the culture dish for long times (weeks) after irradiation and exhibited the ability to secrete growth-promoting factors. 

The significance of the seminal observations reported by Puck and Marcus regarding giant cells was largely overlooked by many research groups, including our own [19], perhaps because such cells cease to proliferate or proliferate at a very slow rate, and thus are often misrepresented as “dead” in the colony formation and other widely used radiosensitivity and chemosensitivity assays [46,47]. 

In 2000, however, Erenpreisa et al. [48] and Illedge et al. [49] reported that giant cells that emerge in cultures of Burkitt’s lymphoma cells, in response to ionizing radiation exposure (10 Gy), can undergo depolyploidization and give rise to progeny that enter the mitotic cycle. This depolyploidization process was subsequently shown by these authors to involve key mediators of meiosis, self-renewal, and mitosis (reviewed in [34,50]). 

As mentioned above, in 2002, Kirsten Walen [3] demonstrated that giant human cells can also give rise to progeny via nuclear budding and bursting (amitosis). In 2004, Rengaswami Rajaraman and his associates [4] reported a similar observation based on live-imaging studies and coined the term “neosis” for this parasexual somatic reduction process of cell division. Rapidly proliferating cells that emerged from giant cells were called “karyoplasts” [3] or “Raju” cells [4] and were shown to exhibit stem cell-like properties. 

In 2004, Navolanic and associates discussed the potential implications of these discoveries in a commentary entitled “Neosis and its Potential Role in Cancer Development and Chemoresistance” [51]. These authors stated that “cell division in eukaryotes by any other process than mitosis or meiosis is certainly a radical concept. Justifiably, the prospect of neosis will be regarded with skepticism unless the claims of the authors can be verified in other laboratories” [51]. In the past decade, numerous groups have indeed verified neosis in different human cell types (e.g., [2,52,53,54,55,56,57,58,59,60,61,62,63]).

### 3.2. Amitosis and System-Level Information Alteration

In the past decade, several reviews have discussed the relationship between genome chaos, new system creation, and evolution [33,64,65,66,67,68]. These include an article entitled “The New Era of Cancer Cytogenetics and Cytogenomics” that highlights the importance of profiling the karyotype (rather than solely profiling gene mutations) especially in cancer, where karyotype alterations contribute to cellular macroevolution [67]. The link between amitotic cell division and the concept of karyotype coding and other means of system-level information alteration remains to be determined.

## 4. Targeting PGCCs in Cancer Therapy

The mechanisms of the formation and fate of PGCCs as well as their prevalence and prognostic value in different types of solid tumors are now well established and extensively discussed [34,35,36,37,38,39,40,41]. Some recent discoveries that suggest the potential of targeting PGCCs for improving patient outcomes are outlined below.

The RhoA-Rock1 pathway, vimentin filaments, and overall actin cytoskeletal network drive the increased stiffness and migratory persistence of PGCCs [2,61,69].The formation of PGCCs and their tumor repopulating progeny (via neosis) can be blocked by the contraceptive drug mifepristone [70].PGCCs developed following cisplatin treatment have a high content of mitochondria and a distinct metabolic profile, which includes high levels of lipid droplets and cholesterol. These PGCCs could be targeted using zoledronic acid, a potent inhibitor of osteoclasts (multinucleated bone cells) [71], which is commonly used to treat osteoporosis and bone metastases [72,73].Treatment with LCL521 or simvastatin disrupts cholesterol signaling and interferes with PGCC progeny formation [74].The sphingolipid enzyme acid ceramidase (ASAH1) is required for the generation of progeny from PGCCs [54,56], and this process (neosis) can be inhibited by tamoxifen, which exerts an off-target effect on ASAH1 [56].Treatment with UC2288 reduces acid ceramide expression and inhibits depolyploidization of PGCCs [75]. UC2288 is an attenuator of p21^WAF1^ (p21) [76], which also exerts off-target effects on the EGFR/ERK (epidermal growth factor receptor/extracellular signal-regulated kinase) signaling pathway [77].The ESCRT (endosomal complexes required for transport) proteins are involved in the budding of PGCCs [78]. Treatment of PGCCs with interferon, a modulator of ESCRT, prevented PGCC budding [78]. In that study, PGCCs were created following exposure to ionizing radiation and were referred to as “radiation-tolerant persister” cells.The expression of PLN4, a perilipin involved in lipid droplet stability, contributes to resistance to cytotoxic therapy in some patients with triple negative breast cancer [79]. PLN4-expressing PGCCs are enriched in tumors of these patients [79].

Some of these discoveries and their clinical implications were discussed in recent blogs entitled “Study discovers potential target for treating aggressive cancer cells” by Kerry Benson [1], “Targeting monster cancer cells could reduce recurrence rates after cancer therapy” by Emma Funk [80], and “First study on physical properties of giant cancer cells may inform new treatments…” by Mollie Rappe [81]. 

Disappointingly, most of these recent reports (including blogs) did not acknowledge the fact that Walen, Rajaraman and others demonstrated the significance of polyploidy-amitosis-neosis in tumorigenesis/therapy resistance about two decades ago. 

Time will tell whether these discoveries will lead to therapeutic strategies that can be adapted in the clinic to improve the outcome of treating patients with solid tumors. To this end, it is important to note that PGCCs represent only one piece of the complex puzzle of therapy resistance and relapse [22,46].

## 5. The Role of PGCCs in Minimal Residual Disease and Tumor Repopulation Post-Therapy

Preclinical studies have identified four phases of solid tumor therapy following exposure to ionizing radiation (e.g., [78]) and chemotherapeutic drugs (e.g., cisplatin [82,83]) that are consistent with common trends in patient outcomes [84]. These phases are outlined below. Figure 3 is adapted from a comprehensive review on polyploidy/neosis by Zhao et al. [85] and a recent report by that group [78] in which they determined the basis for the minimal residual disease (MRD) in the course of radiotherapy. The cartoons showing cell images are redrawn from the time-lapse microscopy data presented in the latter article (see Figure 3B in [78]).

i.*Treatment Phase.* This first phase involves conventional therapeutic strategies (such as surgery, chemotherapy, and radiation therapy) to inhibit tumor growth and hopefully to prevent or at least mitigate metastasis. The majority (perhaps >95%) of anticancer studies have focused on this vital phase of cancer therapy.ii.*Response.* A proportion of cancer cells within a solid tumor responds to genotoxic insult (incurred from the treatment phase) by entering a state of dormancy (active sleep). This group includes PGCCs. In tissue culture studies, it may take ~10 days after treatment for PGCCs to be fully manifested (see, e.g., [78]). It is worth noting that ubiquitously used preclinical anticancer end-points (e.g., multiwell plate cell “viability” and tumor growth delay assays in live animals) are performed within this time frame, and thus they often overlook the impact of PGCCs or score them as “dead” (see, e.g., [22,26]).iii.*Dormancy*. Senescence-like dormancy [84] (also referred to as MRD [85]) is an extended latency period during which PGCCs undergo depolyploidization through a meiosis-like process as well as through amitosis-neosis. PGCCs are also known to have the ability to transfer a small portion of their nuclear material containing stem cell markers to neighboring cells via cytoplasmic tunnels [86].iv.*Recurrence*. Rapidly proliferating progenitor cells emerging from PGCCs repopulate the tumor. Such cells have stem cell characteristics and can show resistance to conventional therapies (used in the “treatment phase”). Whether cells subjected to horizontal gene transfer also exhibit these properties is not known.

The take home message is loud and clear: the initial solid tumor and recurrent tumor post-therapy are different “diseases” requiring different treatment strategies. 

## 6. Amitotic Cell Division Contributes to Intratumor Heterogeneity

Intratumor heterogeneity refers to the presence of different subpopulations of cancer cells within a solid tumor/tumor-derived cell line that exhibit therapy resistance via various molecular and cellular processes. In the 2000s, the presence of cancer stem cells within a solid tumor was emerging as a key contributor to this heterogeneity. As we pointed out in the late 2000s [42,43,44], “another prominent source of cancer cell heterogeneity is via karyoplasts emerging from polyploid/multinucleated giant cells through neotic cell division...Like cancer stem cells, cancer karyoplasts exhibit a high degree of resistance to the cytotoxic effects of cancer therapeutic agents” [42]. Since then, a number of other genetic and non-genetic processes have been demonstrated to contribute to this heterogeneity. These include: (i) reversible dormancy via premature senescence [87,88,89,90]; (ii) reversible dormancy reflecting drug-tolerant persister cells, both preexisting and therapy induced [91,92]; (iii) cancer cell survival subsequent to engagement of apoptosis and other regulated cell death pathways [22,91,93]; and (iv) heterogeneity of p53 protein expression, which “may represent p53 mutant patches indicative of clonal expansion, epigenetic modifications, and/or a number of other possibilities” [94]. 

## 7. Human Genetic Disorders Associated with Genome Instability and Cancer Predisposition: Does Amitosis-Neosis Play a Role?

In the aforementioned review on the “DNA damage repair journey” by Huang and Zhou [15], the authors provided a brief history of xeroderma pigmentosum, an autosomal recessive cancer-prone disorder characterized by DNA repair deficiency. Table 1 contains some clinical and molecular features of this and other extensively studied human hereditary disorders, all of which are characterized by genome instability (for details, see [43]). The relationship (if any) between amitosis/neosis and some clinical features (e.g., cancer proneness) of these disorders remains unknown.

Our group has contributed to the understanding of the DNA damage response of some of these cancer-prone disorders (reviewed in [95]). Of particular relevance to the current discussion, we have observed a high frequency of polyploid/multinucleated giant cells in dermal fibroblast strains from affected members of a cancer-prone family with the Li-Fraumeni syndrome (LFS) [96]. Giant cells were enriched in LFS fibroblasts as a function of in vitro culture age, peaking at passage 38 (Figure 4), or in young (low passage) cultures after exposure to ionizing radiation. We also provided evidence suggesting that the p16^INK4A^ tumor suppressor might play a key role in preventing the genesis of metastatic progeny from polyploid/multinucleated giant cells (i.e., by preventing neosis) [96]. 

Unfortunately, we were unable to secure funds to explore this intriguing hypothesis because, in those days (prior to 2014 [31]), PGCCs were generally considered to reflect artifacts that may only occur in cell lines maintained in culture. In addition, as recently pointed out by Pienta et al. [36], “the majority of cancer research and treatment development communities have disregarded these (giant) cells as irreversibly senescent or destined for mitotic catastrophe and death.” 

The significant roles played by giant cells in tumorigenesis and therapy resistance are becoming increasingly appreciated, which serves to underscore the pioneering contributions of Walen and Rajaraman to the polyploidy-amitosis-neosis field about two decades ago.

## 8. Conclusions

Studies reported since the 1950s have revealed that moderate, clinically relevant doses of ionizing radiation and chemotherapeutic drugs trigger the generation of cells with various manifestations of genome chaos, including PGCCs, that remain viable, secrete growth-promoting factors, and are capable of producing progeny with stem cell-like properties. PGCCs can contribute to tumor repopulation by various means, including: (i) depolyploidization, through which PGCCs reduce their genome to a size (paradiploid) that is compatible with mitosis; (ii) amitotic cell division, which results in the emergence of numerous (hundreds to thousands) of proliferating daughter cells from a single PGCC; and (iii) horizontal gene transfer via cytoplasmic tunnels by which PGCCs can export a small portion of their nuclear material containing stem cell markers to neighboring cells (reviewed in, e.g., [26,46]).

Surprisingly, PGCCs continue to be overlooked in most cancer-related articles that discuss the “bright” (pro-death) side of apoptosis in solid tumor therapy [97,98,99,100,101], the “dark” (pro-survival) side of apoptosis in cancer therapy [91,93], targeting the DNA damage response (e.g., the p53/p21 pathway) in cancer therapy [102,103,104,105,106,107], as well as numerous articles that highlight limitations of implementing anticancer strategies under the term “precision oncology” (reviewed in [22,26]).

When discussing the possible reasons for this serious oversight (disregarding PGCCs) with the senior author of a comprehensive review on breast cancer, he replied in an email communication that “we are all learning… According to the 19th century German philosopher Arthur Schopenhauer, All truth passes through three stages: First, it is ridiculed. Second, it is violently opposed. Third, it is accepted as self-evident”.

The lessons that we have learned from landmark discoveries of Walen, Rajaraman, Erenpreisa, and other pioneering PGCC biologists are invaluable, and will hopefully lead to improved efficacy of conventional solid tumor therapies.

## Figures and Tables

**Figure 1 cancers-16-03106-f001:**
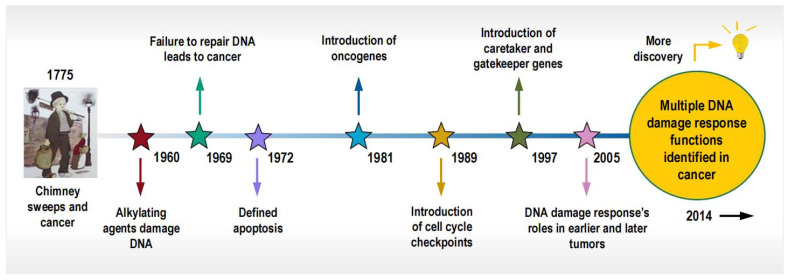
Timeline of findings related to the DNA damage response that led to the somatic mutation theory of cancer. Reproduced from Huang and Zhou [15].

**Figure 2 cancers-16-03106-f002:**
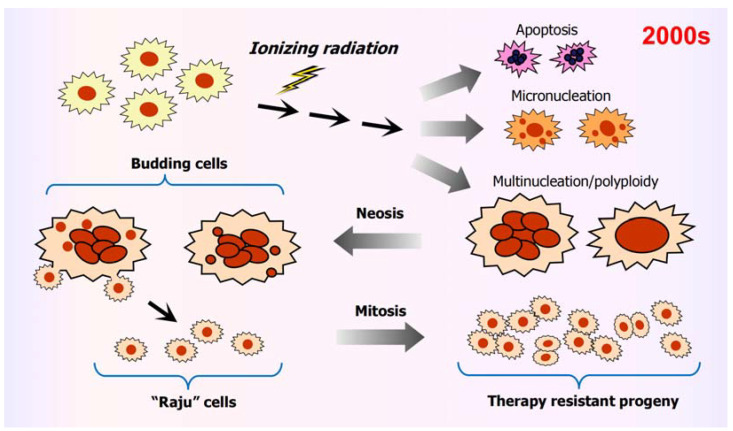
Fate of giant cancer cells based on discoveries reported by Walen, Rajaraman, Erenpreisa and their colleagues in the early 2000s (reviewed in [42]). These authors revealed the following sequence of events: “Cancer cells exposed to DNA-damaging agents may fail to activate early cell cycle checkpoints and thus replicate their genome on damaged templates. While some cells may die, a great number may retain viability and acquire diverse types of nuclear abnormalities including micronuclei, multiple nuclei and a highly enlarged nucleus. A proportion of multinucleated/polyploid giant cells may undergo neotic cell division, which is characterized by karyokinesis via nuclear budding followed by asymmetric cytokinesis, resulting in the generation of small mononuclear karyoplasts (also called Raju cells). These karyoplasts may resume mitotic division, and may eventually produce highly metastatic and therapy resistant descendants” [42]. Reproduced from our 2008 review [42].

**Figure 3 cancers-16-03106-f003:**
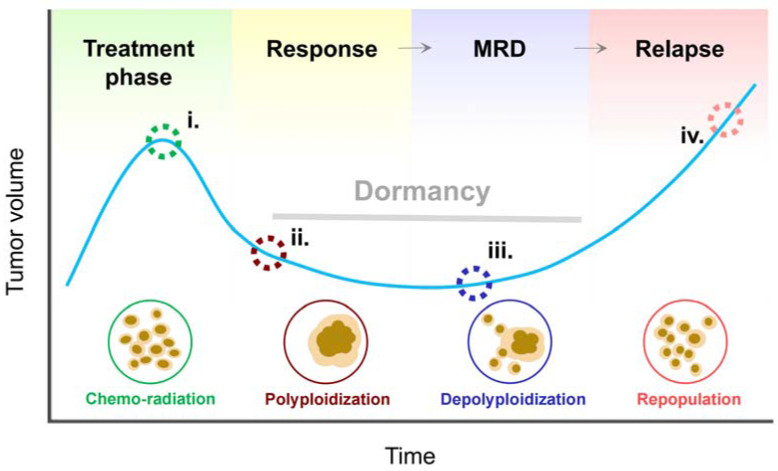
The role of PGCCs in cancer remission (dormancy) and relapse. At the macroscopic level, (e.g., observed in live animals [82,83]), the initial cancer remission post-therapy is followed by a dormant state (also known as “senescence-like dormancy” [84] and “minimal residual disease” [85]), and eventual tumor relapse in most patients. At the microscopic level, in response to ionizing radiation [78] or chemotherapeutic drugs [82,83] (i, green circles), the majority of bulk (para-diploid) cancer cells undergo polyploidization and/or cell fusion, generating PGCCs (ii, maroon circles). During the subsequent transitional state (iii, blue circles), a proportion of PGCCs undergo depolyploidization and generate high frequencies of rapidly proliferating small-sized progeny cells that repopulate the tumor (iv, pink circles).

**Figure 4 cancers-16-03106-f004:**
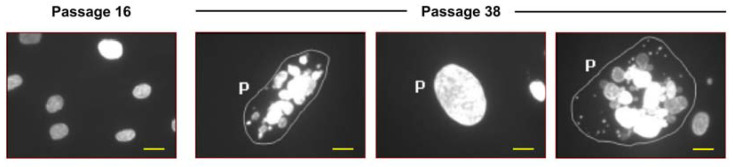
Fluorescence images showing the nuclear morphology of 2800T (Li-Fraumeni syndrome) dermal fibroblasts at different passage numbers. P, polyploid/multinucleated giant cells. Scale bars, 50 µm. Reproduced from Mirzayans et al. [96] with permission.

**Table 1 cancers-16-03106-t001:** Clinical and molecular characteristics of the indicated human cancer-prone disorders. For details, see [43].

Disorder	Mode of Transmission	Defective Protein	Defective Function
Xeroderma pigmentosum	Autosomal recessive	XPA through XPG, DNA polymerase η	Nucleotide excision repair (XPA through XPG), postreplication repair (XPV)
Ataxia telangiectasia	Autosomal recessive	ATM	ATM signaling
Li-Fraumeni syndrome	Autosomal dominant	p53; Chk2	p53/Chk2 signaling
Nijmegen breakage syndrome	Autosomal recessive	NBS1	DSB repair
Werner syndrome	Autosomal recessive	WRN	DNA helicase
Bloom syndrome	Autosomal recessive	BS	DNA helicase
Rothmund-Thompson syndrome	Autosomal recessive	RTS	DNA helicase
Fanconi anemia	Autosomal recessive	FANCA, B, C, D1, D2, E, F, G, I, J, L and M	DNA helicase, DNA cross-link repair

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
