# Peer review of "Amitotic Cell Division, Malignancy, and Resistance to Anticancer Agents: A Tribute to Drs. Walen and Rajaraman"

_cancers, 2024, doi:10.3390/cancers16173106_

Round 1

Reviewer 1 Report

Comments and Suggestions for Authors

The present research constitutes a comprehensive investigation into a crucial topic, comprising a meticulous exposition and analysis of accumulated data. Scholars lay the foundation for vigorous advancement in the notion that chromosomal instability has a profound influence on tumorigenesis.

Comments on the Quality of English Language

The present research constitutes a comprehensive investigation into a crucial topic, comprising a meticulous exposition and analysis of accumulated data. Scholars lay the foundation for vigorous advancement in the notion that chromosomal instability has a profound influence on tumorigenesis.

Author Response

Thank you!

Reviewer 2 Report

Comments and Suggestions for Authors

Author Response

We thank this reviewer for considering our commentary to be “mostly well-written review at the cutting edge of the studies of many discoveries on cancer resistance to drugs involving reversible polyploidy of induced polyploid giant cells (PGCCs) updated to 2024…”

We have made revisions to incorportae most of the suggestions by this reviewer in order to improve the precision of the contribution of Rajaraman and Walen and to ordering the entangled reference numeration.  

Reviewer 3 Report

Comments and Suggestions for Authors

Thank you for the opportunity to review your Tribute to Drs. Walen and Rajaraman. It was incredibly interesting and thorough. Excellent use of diagrams, tables and bullet points to make the information more accessible. 

My only suggestion would be the 'Food for Thought' section regarding the potential use of the Newcastle Disease virus could use a little expanding... it currently seems a big idea to tack on at the end.

Author Response

Thank you for your comments. I saw them when I was uploading the revised manuscript.

I agree that the material under "Food for Thought" is of particular significance, but unfortunately it was out of place. We decided to delete it and instead expand the text and publish as a separate commentray.

Acoordingly, the curremt paper ends with a statement on the contributions of Walen and Rajaraman.

Reviewer 4 Report

Comments and Suggestions for Authors

This commentary presents a debatable opinion. polyploidy cells may perform amitosis and are related to tumorigenesis and metastasis. Although there are some indirect evidence support it but never presents solid direct evidence. With the development of knowledge, this conclusion may be confirmed. This commentary is well-organized and presented logically. Different opinions should be accepted by the journals.

Author Response

We thank this reviewer for considering this commentary to be well-organized and presented logically. Like other journals, cancers do publish other hypotheses, but the puropse of this commnetay is to highligh that amitosis of polyploid cells that is becoming increasingly appreciated (e.g., as discussed in the 2024 Advanced Science review by Zhao et al. [85]), was first reported by Walen and Rajaraman over 20 years ago based on solid experimental data rather than hypotheses or opinions. This knowledge, which began accumulating since 1956 [45], has been confirmed by numerous solid expecrimental and clinincal data, as extensively reviewed by us and others (see, e.g., [46] and references therein).